# Cell-Cycle Dysregulation in the Pathogenesis of Diabetic Kidney Disease: An Update

**DOI:** 10.3390/ijms24032133

**Published:** 2023-01-21

**Authors:** Bowen Deng, Anni Song, Chun Zhang

**Affiliations:** Department of Nephrology, Union Hospital, Tongji Medical College, Huazhong University of Science and Technology, Wuhan 430022, China

**Keywords:** diabetic kidney disease, renal fibrosis, cell-cycle arrest, overproliferation, inflammation

## Abstract

In the last few decades, the prevalence of diabetes mellitus (DM) has increased rapidly. Diabetic kidney disease (DKD) is the major cause of end-stage renal disease (ESRD) globally, attributed to hemodynamic changes and chronic hyperglycemia. Recent findings have emphasized the role of cell-cycle dysregulation in renal fibrosis and ESRD. Under normal physiological conditions, most mature renal cells are arrested in the G0 phase of the cell cycle, with a rather low rate of renewal. However, renal cells can bypass restriction points and re-enter the cell cycle under stimulation of injuries induced via metabolic disorders. Mild injuries activate proliferation of renal cells to compensate for cell loss and reinstate renal function, while severe or repeated injuries will lead to DNA damage and maladaptive repair which ultimately results in cell-cycle arrest or overproliferation, and eventually promote renal fibrosis and ESRD. In this review, we focus on the role of cell-cycle dysregulation in DKD and discuss new, emerging pathways that are implicated in the process.

## 1. Introduction

In the past few decades, the population living with diabetes mellitus (DM) has increased dramatically. As of 2021, there are 537 million adults with diabetes in the whole world, and that number is predicted to increase to 783 million by 2045 [1]. DM is a long-term disease that changes the metabolic milieus of its patients. If poorly controlled, patients with DM will suffer from its complications of multiple systems, such as hypertension, myocardial infarction (MI), and proteinuria, in the long run. In patients with diabetes, diabetic kidney disease (DKD) is a frequent and severe complication that has an increased morbidity and mortality rate [2]. DKD refers to the presence of altered kidney function in diabetic patients who do not have any other cause of chronic kidney disease. Diagnosis of DKD is based on decreased estimated glomerular filtration rate (eGFR < 60 mL/min/1.73 m^2^) and/or increased urinary albumin excretion (≥30 mg/g creatinine) for >3 months, according to the American Diabetes Association’s latest guidelines [3]. It is well-accepted that as a consequence of DKD, kidney size increases, mainly because of a buildup of extracellular matrix proteins [4]. Glomerular lesions are regarded as characteristic and decisive changes in the biopsies of patients with either type 1 DM (T1DM) or type 2 diabetes mellitus (T2DM), featuring diffuse and nodular mesangial expansion and glomerular basement membrane (GBM) thickening as well as arteriolar hyalinosis [5]. 

Cell-cycle dysregulation refers to the situation where quiescent cells bypass checkpoints and re-enter the cell cycle, or where proliferative cells are arrested in a certain phase of the cell cycle. Tight control of the cell cycle maintains cell and tissue homeostasis and ensures that physiological activities are precisely performed. Regulation of the cell cycle is driven and performed by cyclin-dependent kinases (CDKs), cyclins, and cyclin-dependent kinase inhibitors (CKIs) [6]. Generally, most renal cells are differentiated cells that have stopped cell division and entered the quiescent stage, which is known as the G0 phase (Figure 1). Cell-cycle dysregulation happens when renal cells are suffering from injuries and metabolic disorders caused by DM; they can be activated and initiate the proliferation process to compensate for loss of renal function and repair or be arrested in the G2/M phase and initiate cell loss and renal fibrosis [7]. Under pathological conditions that usually come after severe injuries, the cell cycle of renal cells is prone to manifesting aberrant patterns, either excessive proliferation or re-entering of the G0 phase permanently, resulting in pathological changes in the kidney. Such processes consequently prompt glomerular and tubulointerstitial fibrosis and eventually chronic kidney disease (CKD) [8]. Meanwhile, the medical needs for DKD remain unmet [5], and this provides a therapeutic target for people to research. This process involves each part of the kidney; in this review, we will look into concrete mechanisms of DKD, respectively in glomerular cells, tubular cells, and other cells that promote this process, such as inflammatory cells.

## 2. Interactions between Cell-Cycle Dysregulation and Other Cellular Processes

In the kidney, hyperglycemia arouses multiple biochemical pathways, such as upregulation of advanced glycosylation end products (AGEs), increased mitochondrial flux, and PKC, leading to DNA damage, upgraded growth factors, and inflammatory cytokines and eventually resulting in podocyte damage and loss, excessive mesangial-matrix deposition, glomerulosclerosis, and tubular fibrosis. Multiple pathophysiological processes, including podocyte injury, altered tubuloglomerular feedback, inflammation, mitochondrial dysfunction, and impaired autophagy, are established following the above events [9]. Excess energy and upgraded mitochondrial flux transform the balance of mitochondrial homeostasis into mitochondrial fission, mediated with ROCK1 activation in podocytes and endothelial cells [10], resulting in increased injured and dysfunctional mitochondria and promotion of oxidative stress and inflammation [11]. Since renal tubular epithelial cells are enriched with mitochondria, they are vulnerable to mitochondrial dysfunction.

Autophagy is an essential cell-recycling process that eliminates damaged organelles and proteins using proteolytic enzymes within lysosomes, reuses energy, and renews the cell population. Normally performed autophagy eliminates DNA-damaged cells and cells with perturbed cell cycles, while impaired autophagy contributes to progression of DKD [9]. On one hand, podocytes and proximal tubular cells exposed to high glucose concentrations manifest as impaired autophagy [12]; on the other hand, mice with podocyte-specific knockouts of key autophagy molecules suffer from proteinuria, glomerulosclerosis, and kidney failure [13]. There is an association between podocyte autophagy and mesangial expansion reduction, improved glomerular histology, and decreased albuminuria. Metformin prevents progression of DKD, independent of its glucose-lowering effects, via induction of MC autophagy [14]. A study showed that the targets of rapamycin (TOR)–autophagy spatial coupling compartments (TASCCs) were formed in G2-M TECs, which promoted production of profibrotic cytokines [15], indicating crosstalk between autophagy and cell-cycle dysregulation.

## 3. Glomeruli and Diabetic Kidney Disease

As aforementioned, the pathological changes of DKD are highly associated with glomerular injury. GFR elevation occurs in the early stages of diabetes, implying histopathological changes in glomeruli [16]. The glomerular filtration barrier consists of glomerular endothelial cells (GECs), the glomerular basement membrane (GBM), and podocytes (Figure 2). The earliest observable glomerular change with light microscopy was diffuse mesangial expansion, which occurs as early as the fifth year after the onset of T1DM [17]. As the disease progresses, diffuse mesangial expansion proceeds to nodular lesions and eventually results in nodular mesangial expansion. The earliest change that can be observed with electron microscopy (EM) is GBM thickening, which occurs between 2 and 8 years after the onset of diabetes [18].

### 3.1. Cell-Cycle Dysregulation of Podocytes

Apart from diffuse mesangial expansion and GBM, which are most commonly found in biopsies, a cell analysis of samples from DM patients suggests that podocyte number could be the best predictor of disease, and podocyte loss has close contact with proteinuria [19]. Podocytes are highly differentiated pericyte-like cells that cover the basement membranes of glomeruli with their foot processes. Their subtle and delicate cytoarchitecture enables them to participate in the composition of the glomerular filtration barrier, which prevents urine protein from leaking; thereby, injury or loss of podocytes can lead to both proteinuria and nephrotic syndrome, which make up the major causes of CKD [20]. However, with stimulation of hyperglycemia, podocytes can re-enter the cell cycle and end up with mitotic catastrophe.

In the early stages, precursor cells for podocytes proliferate in an immature and undifferentiated state. In this state, proliferating cell nuclear antigen (PCNA) and Ki-67, as well as cyclin A (required for DNA synthesis) and cyclins B1 and Cdc2 (required for mitosis), are expressed and detectable, while CKIs such as p21^Cip1^, p27^Kip1,^ and p57^Kip2^ are lacking [21]. p21^Cip1^ and p27^Kip1^ are found to be upgraded in glomerular sclerosis, with increased glomerular volume [22]. In mature kidneys, there is little turnover of adult kidney cells under physiological conditions because mature podocytes upgrade levels of CKIs and reduce expressions of cyclins and CDKs, and a high level of CKIs will probably result in renewal of podocyte disability [7]. Studies have shown that when exposed to injury, podocytes will possibly return to the cell cycle and proliferate, and thus their number would increase. In certain diseases, the differentiated phenotype is dysregulated, such as in idiopathic collapsing glomerulopathy, HIV-associated nephropathy, and crescentic glomerulonephritis. This can be indicated through loss of maturity markers and proven through absent CKIs and upgraded cyclin D1 and other proliferative proteins [21]. However, podocytes in diseases such as minimal change disease, FSGS, membranous nephropathy, and DKD lack the ability to proliferate and stay arrested in the cell cycle, even in pathological conditions. Even though a study has shown that when exposed to high glucose levels, most podocytes entered the S and G2/M phases, which facilitated the mitotic catastrophe process [23], evidence of mature podocyte proliferation is lacking. Diabetic kidney disease is featured with increased GFR and ensuing high podocyte shear stress, and these conditions make podocytopathy likely to develop [20]. Podocytes cover the filtration barrier with their specified foot processes and sustain the hydrostatic pressure difference across the filtration barrier. With higher levels of glucose in a DM patient’s proximal tubular, sodium/glucose cotransporters (SGLTs) are more occupied than those in normal conditions and reabsorb more sodium than usual, and the reduced levels of sodium will in turn activate the renin–angiotensin–aldosterone system (RAAS), resulting in afferent arteriole vasodilatation, efferent arteriole vasoconstriction, and eventually an increased single-nephron GFR through tubuloglomerular feedback [16]. This increased GFR and high podocyte shear stress push podocytes to detach from the GBM and prevent their cell proliferation [24].

### 3.2. Cell-Cycle Dysregulation of Mesangial Cells

Mesangium is made up of the Greek word mesos, meaning middle, and the Latin suffix angium, meaning container, and literally refers to the middle space or the middle vessel [25]. Mesangial cells promote glomerular development, produce the glomerular basement membrane matrix, and structurally support glomerular capillaries through counteraction of the hydrostatic pressure gradient across the filtration barrier [25]. When activated through injury, the mesangial cells activate to recruit immune cells, control inflammation, and function in a reparative way in response to injury. In normal physiological conditions, cell cycle inhibitor p27 in mature mesangial cells (MCs) is upregulated, and these cells are arrested in the G0 phase. Quiescent MCs can be stimulated to proliferate via all kinds of injury stimulation, resulting in increased numbers of MCs and persistent cellular-matrix accumulation. It has been proven that in models of mesangial proliferative glomerulonephritis and Thy1 nephritis, proliferative markers, such as cyclin D, cyclin E, cyclin A, CDK2, and CDK4, are upregulated [26].

Proliferation of MCs and excessive extracellular matrix (ECM) formation could lead to glomerular fibrosis and glomerulosclerosis [27]. This proliferation can be initiated through stimulation of injury-induced mitogens, featuring p27 downregulation and CDK/cyclin upregulation. According to another study, MC proliferation induced via high glucose was accompanied by decreased p21 protein expression as well as increased CDK4 and CDK2 kinase activities. It has been proven that CDK4 and CDK2 activity could be decreased with simvastatin through enhancement of p21 protein expression, suggesting that statins could inhibit renal fibrosis through inhibition of MC proliferation [7]. The Hippo pathway, which inhibits cell proliferation, is also involved in MC proliferation in DN [28]. Those researchers found that in vitro, proliferation of glomerular MCs was prompted via incubation with high glucose; meanwhile, the Hippo pathway was significantly inactive. It was found both in db/db mice’s glomerular MCs and in high glucose-cultured glomerular MCs that as a result of reduced phosphorylation of MST1 and Lats1, Yes-associated protein (YAP) was expressed and underwent nuclear translocation, and subsequently, the combination of YAP with the TEA/ATS domain (TEAD) was increased and the expressions of downstream target genes such as cyclinE were upregulated, resulting in mesangial cell proliferation and ECM accumulation. However, it was found in another study that high glucose could induce ferroptosis in MCs in vitro [29].

### 3.3. Cell-Cycle Dysregulation of Glomerular Endothelial Cells (GECs)

GECs are a crucial component of the renal filtration barrier, exposed to the blood circulation directly and vulnerable to lipids, inflammatory factors, and hyperglycemia-induced injury caused by diabetes [30]. To handle albumin filtration, GECs are highly fenestrated, with every individual fenestra on the order of 70–100 nm in diameter, which is approximately 17 times larger than the diameter of albumin [30]. Podocytes have been extensively studied as primary targets in DKD, but GEC dysfunction has recently been linked to glomerular sclerosis, including DKD. How GEC cell-cycle dysregulation leads to or aggravates DKD is not completely clear, but it is understood that VEGF mediates angiogenesis and induces glomerular injury through promotion of GEC proliferation.

Vascular endothelial growth factor (VEGF) and its receptors (VEGFR-1 and VEGFR-2) are one of the most intensively investigated regulatory loops of GEC proliferation. In early DKD, both VEGF and VEGFR2 have been shown to be increased in renal tissue [31]. Some findings suggest that proliferation of endothelial cells is regulated with VEGF produced by their adjacent podocytes [32]. The VEGF-A/VEGFR-1 system plays a vital role in development and function of the glomerular filtration barrier in the kidney, while the VEGF-A/VEGFR-2 system is responsible for endothelial cell migration, proliferation, and differentiation [32]. Accordingly, inhibition of VEGF-A or the VEGFRs could alleviate glomerular injuries in diabetic animals [33]. However, in later DKD, production of VEGF is reduced with loss of podocytes, followed by vascular rarefication and renal fibrosis [32]. In such cases, VEGF might be a protective factor of renal fibrosis. It was reported that a sodium–glucose cotransporter 2 (SGLT2) inhibitor, luseogliflozin, could attenuate vessel injuries and renal fibrosis through a VEGF-dependent pathway [34]. It was observed that luseogliflozin significantly increased (VEGF)-A in the kidney after ischemia/reperfusion injury, but not in animals co-treated with sunitinib, a VEGF receptor inhibitor. Although not verified in the diabetic situation, this finding suggests that proliferation of GECs is protective of renal fibrosis.

Other regulatory loops, such as angiopoietins (Ang)/Tie-2 and endothelin-1 (ET-1)/ET_A_, are also implicated in GEC overproliferation and development of DKD. A study showed that Ang-1 overexpression could reduce albuminuria and prevent GEC proliferation via the Ang-1/Tie-2 pathway [35]. Therefore, GEC overproliferation promotes development of early DKD while also laying the groundwork for renal fibrosis in later DKD.

## 4. Kidney Tubules and Diabetic Kidney Disease

DKD is featured with glomerulosclerosis and renal interstitial fibrosis (RIF) [36]. Some have claimed that the best indicator of functional progression within diabetic patients is tubulointerstitial disease, rather than abnormalities in the glomerulus [37]. As a consequence of DKD, tubular injury that occurs in the early stages of DKD is considered more popular than glomerular injury in biopsy analysis. A study showed that in patients with microalbuminuria, only 30% had typical diabetic glomerulopathy, and 40% of them suffered from tubulointerstitial and/or vascular lesions that were more advanced than diabetic glomerulopathy [38]. Increased single-nephron GFR, one of the characteristics of DKD, is now attributed to alterations in tubuloglomerular feedback. Apart from that, tubule hypertrophy also occurs underlying development and progression of DKD [9]. In the early stages of diabetes, with disordered glucose metabolism, proximal tubules reabsorb more glucose due to hyperglycemia, leading to proximal tubule hypertrophy, which is also related to reactive oxygen species production, oxidative damage, and TGF-β production [39]. These processes involve cell-cycle arrest of tubular epithelial cells and overproliferation of fibroblasts.

### 4.1. Cell-Cycle Dysregulation of Tubular Epithelial Cells (TECs)

Both in normal kidneys and in kidneys with diabetic nephropathy, glucose molecules are reabsorbed by sodium–glucose cotransporter 2 expressed on TECs in the proximal tubule after filtration of the glomeruli. Under physiological conditions, these TECs rarely fall off of the renal tubules, with approximately one TEC per human nephron daily [40]. To remedy cell loss and compensate for the reabsorptive capacity of the proximal tubule, TECs in mature mammals keep proliferating at a rather low rate; this was proven with immunostained PCNA and Ki-67 [41]. In fact, most proximal tubular cells, particularly those in the S3 segment, express markers of the G1 phase rather than of the G0 phase [42]. 

However, the dividing rate of TECs could remarkably increase upon stimulation of injury, which indicates their potential ability to proliferate. Studies have shown that high glucose induces excessive production of reactive oxygen species (ROS) and upregulates expression of transforming growth factor-β1 (TGF- β1) in TECs [43]. After stimulation of TGF- β or γ-irradiation in renal epithelial cells, p21 inhibits cell cycle progression at G2/M [44]. Proliferation and division of TECs could be activated even in the early stages of DM. A study showed that albumin could help initiate the proliferation of opossum kidney cells in a culture, which was mediated with PI 3-kinase. This study demonstrated that albuminuria could initiate interstitial inflammation and scarring, relating glomerular endothelium injury to interstitial fibrosis [45]. An in vitro experiment reported that p38 MAPK and NF-κB (which is regulated with p38 MAPK) in proximal tubular epithelial cells (PTECs) are activated via high-glucose cultures, which are implicated in migration and proliferation [46]. Not only does crosstalk amplify inflammatory signaling; it also suggests migration and proliferation of PTECs. Pirfenidone, a TGF-β blocker, was found to antagonize the MAPK pathway to attenuate EMT and fibrosis in animal studies [47].

The RAAS system also stimulates proliferation of TECs. The expression of VEGF was observed to be upgraded in the proximal tubules of mice with type 1 diabetes. In vitro, the mouse PTECs cultured with 30 mM glucose (HG) for 24 h turned out to increase the expression of VEGF at both the protein and mRNA levels. This is because HG stimulates synthesis of angiotensinogen (AGT) and activation of renin and angiotensin-converting enzyme (ACE), resulting in local production of Ang I and Ang II, and the latter stimulates synthesis of VEGF; meanwhile, HG activates the ERK pathway, which in turn upregulates expression of AGT and sustains activation of itself [48]. Another study revealed that in early diabetic kidneys, the activity of ornithine decarboxylase (ODC) was increased via the activation of the JAK/STAT pathway. ODC is the rate-limiting enzyme in the synthesis of polyamine, which is essential for hypertrophy and hyperplasia of early DKD [49].

In fact, TECs and renal interstitial fibroblasts are two major sources of myofibroblasts [50]. Epithelial–mesenchymal transition (EMT) occurs in TECs during the process of renal fibrosis, making TECs express both epithelial and mesenchymal cell markers. EMT can lead to renal fibrosis while inhibition of EMT will mitigate this process [7]. DNA damage caused by injury downregulates CDK1 and cyclin B kinase activity, respectively, through regulation of p53 and CDC25 [51]. The G2/M phase arrest is also associated with renal fibrosis. Under stimulation of severe or persistent damage, TECs can be arrested in the G2/M phase of the cell cycle and halt the process of proliferation. A study showed that HIF-1α upregulated p21 expression and induced G2/M arrest and fibrogenesis in HK-2 cells treated with aristolochic acid, and the inhibition of p21 prevented G2/M phase arrest and fibrogenesis in those cells, suggesting that there is a connection between G2/M phase arrest of PTECs and renal fibrosis, and that p21 modulates the process [52]. Further mechanisms remain to be investigated. Knockout of cyclin G1 and inhibition of 4E-BP1 (a downstream molecule of the mTORC1 pathway) both alleviate the arrest of the G2/M phase [15] and renal fibrosis [53], suggesting a connection between cell-cycle dysregulation and renal fibrosis. 

### 4.2. Cell-Cycle Dysregulation of Fibroblasts in the Renal Interstitium

A consistent finding in diabetic tissues is activation of fibroblasts and increased extracellular matrix synthesis, and ECM accumulation will finally lead to tubulointerstitial fibrosis (TIF). TIF is one of the features of renal fibrosis, the final common pathology of CKD [54]. The extracellular matrix exudates and deposits excessively in the latent space outside the renal tubules, facilitating wound healing and resulting in renal fibrosis. This attributes mainly to secretion of myofibroblasts, which are generally considered the major extracellular matrix producers. There are five different sources of myofibroblasts in fibrotic kidneys of mammals (Figure 3): activation of interstitial fibroblasts, differentiation of pericytes, translation of tubular epithelial cells, endothelial cells, and recruitment of circulating fibrocytes. Myofibroblasts are mainly derived from activated resident fibroblasts [55], and it has been reported that glycated collagen can promote conversion from fibroblasts to myofibroblasts [56].

Activation of fibroblasts consists of two characteristics: proliferation and translation to myofibroblasts that produce ECM and express mesenchymal marker α smooth muscle actin (α-SMA). Under normal physiological conditions, renal fibroblasts are arrested in the G0 phase and keep quiescent. However, upon injury and with stimulation of cytokines, these fibroblasts can re-enter the cell cycle and proliferate. Growth factors such as PDGF, TGF-β, FGF2, and CTGF are also well-known mitogens that promote fibroblast overproliferation. TXNDC5 (endoplasmic reticulum protein) and SIRT2 (sirtuin 2) both promote renal fibrosis through upregulation of TGF-β1 expression in kidney fibroblasts [57]. A high level of mitochondrial fission is associated with cancer or organ fibrosis that promotes proliferation. Phosphorylation of Drp1 (dynamin-related protein 1) at serine 616 was found to increase in a mouse model of obstructive nephropathy; in cultured renal interstitial fibroblasts, a block of Drp1 suppressed the activation and proliferation mediated with TGF-β1, indicating a new pathway of fibroblast proliferation [58]. There is also crosstalk between TECs and fibroblasts. It was revealed that the miR-21/PTEN/Akt pathway activates fibroblasts via exosomal miR-21 from TECs, which can accelerate development of renal fibrosis [59]. However, more detailed mechanisms about how injuries trigger activation and proliferation of fibroblasts remain unclear. 

## 5. Pericytes of Renal Microvessels and Diabetic Kidney Disease

DM is a chronic systematic disease that changes the metabolic milieus of its patients and affects multiple systems. Prolonged and chronic exposure to hyperglycemia leads to microvascular injuries, resulting in retinopathy, neuropathy, and nephrology. In fact, kidneys are vulnerable to the effects of hyperglycemia due to their substantial distribution of capillary loops, and microvascular complications are the major characteristic of DKD. Pericytes are vascular mural cells derived from mesenchymal origins and are well-known for their regulation of vascular structure. Under normal physiological conditions, pericytes appose to vascular endothelial cells, including peritubular capillaries of the kidney, and regulate vessel stabilization and function. The absence of vascular smooth muscle cells (VSMCs) in capillaries led to a hypothesis that pericytes interact with endothelial cells and function as VSMCs [60].

Unlike highly differentiated cells (e.g., podocytes and mesangial cells), pericytes possess the potential capacity for proliferation and differentiation, which enables them to differentiate and proliferate. The capillary basement membrane between pericytes and endothelial cells is frequently incomplete, and pericytes could reach into the gaps of endothelial cells, enabling interaction between pericytes and endothelial cells. This interaction maintains the stability of kidney peritubular microvasculature under normal conditions, initiates vascular sprouting and leads to completely differentiated vasculature through various signaling pathways after injury [60]. Platelet-derived growth factor (PDGF)-BB secreted from endothelial cells binds to its receptor, PDGF-β, expressed on the membrane of pericytes, initiating pericyte proliferation and migration [61]. Angiogenesis factors, such as VEGF, promote pericyte detachment from the basement membrane through stimulation of endothelial cells prior to proliferation [62]. The communication between endothelial cells and pericytes is impaired in the diabetic milieu, which emphasizes the part that cell-cycle dysregulation of pericytes plays in diabetic vasculopathy [63]. However, the specific mechanisms thereof are still by and large elusive. 

There is increasing evidence that peritubular capillary pericytes are the main source of scar-forming myofibroblasts in chronic kidney disease (CKD) and acute kidney injury (AKI). Using single-cell RNA-seq, a study revealed that human kidney fibrosis is characterized with distinct subpopulations of pericytes and fibroblasts that cause scarring [55]. In the absence of specific cellular markers and strategies to trace lineages, researchers never made it to identify pericytes and their contribution to renal fibrogenesis, since pericytes and fibroblasts are both distributed in the subendothelial region and have the same cellular markers [64]. It is true that the markers for pericytes used to indicate pericyte–myofibroblast conversions in several reports do not always identify pericytes and are present in a variety of cells, including interstitial fibroblasts. In 2015, a Gli1+ subpopulation of pericytes was identified as myofibroblast progenitors. That study showed that Gli+ cells proliferate in the interstitial area and attain expression of NG2 and α-SMA, the same as myofibroblasts [65]. The study also revealed the mechanism of renal fibrosis in which cell-cycle dysregulation of pericytes was involved. However, further investigation is still needed to consummate cognition of the mechanisms.

## 6. Other Cells in Diabetic Kidney Disease

Apart from the residential cells mentioned above, inflammatory cells also participate in progression of DKD and promote fibrogenesis of kidneys, including glomerulosclerosis and interstitial sclerosis. Fibrosis is fundamentally a process of healing and repair after injury, maintaining the original organ architecture and ensuring its general functions. Repair of tissue requires inflammation, with the exception of embryos, where tissue can be repaired without the usual inflammation [66]. Systemic and local renal inflammation can be caused by diverse stimuli, such as macrophages; nuclear transcription factor-kappa B (NFκB); the Janus kinase/signal transducer and activator of transcription (JAK/STAT) pathway; inflammatory cytokines; and other key inflammatory cells, molecules, and diverse transduction pathways. The inflammatory processes were proven to correlate to DKD and renal inflammation. An increase in macrophage infiltration or mast cell degranulation relates to a decrease in the estimated glomerular filtration rate, indicating the role of inflammatory cells in DKD [67]. 

These inflammatory cells affect diabetic kidney disease and promote organ fibrosis not through their proliferation but through mediation of cell-cycle dysregulation of residential cells. In fact, almost all types of interstitial cell, including residential fibroblasts, TECs, VSMCs, and a subpopulation of macrophages, have the potential to be activated and produce ECM, which leads to renal fibrogenesis. This process is greatly influenced by transforming growth factor-beta profibrotic cytokines (TGF-βs) and can be made inverse with inhibition of TGF-βs or their downstream pathways [68]. Under the conditions of diabetes milieus, hyperglycemia can trigger macrophages to release senescence-associated secretory phenotype (SASP) components, causing low-grade inflammation in order to promote cellular senescence of MCs and TECs [39]. 

Kidney injury molecule (KIM-1), an immunoglobulin superfamily protein, is significantly upregulated in the proximal tubules of injured kidneys and regarded as the biomarker for human renal proximal tubule injury [69]. It was shown in a study that KIM-1 mediates endocytic uptake of palmitic acid (PA)-bound albumin, and KIM-1-mediated PA-albumin uptake leads to ensuing DNA damage, interstitial inflammation, and fibrosis [37]. During treatment with PA-BSA, there were more inflammatory cells surrounding the proximal tubules in wild-type mice than in KIM-1^Δmucin^ mice whose mucin domain of KIM-1 was deleted. In addition, immunofluorescence showed that macrophage F4/80-positive staining was more prevalent in the interstitia of wild-type diabetic mouse kidneys than those of KIM-1^Δmucin^ diabetic mouse kidneys, especially around proximal tubules that expressed KIM-1, suggesting a pathway of inflammation and fibrogenesis. This study also showed that PA-albumin could induce DDR and lead to G2/M cell-cycle arrest of proximal cells, which was identified to contribute to development of tubulointerstitial inflammation and scarring in an earlier study [70]. Interestingly, KIM-1 has been proven a receptor for SARS-CoV-2 in the lung and the kidney and is responsible for acute kidney injury following SARS-CoV-2 infection through mediation of internalization of PA-bound albumin as well [71]. 

## 7. Conclusions and Perspectives

The strictly controlled cell cycle of mammalian cells is essential for the homeostasis and development of both the cell and the creature, and it is vital for intact renal functions. It is tightly regulated with cell-cycle regulators, including CDKs, cyclins, and CKIs. Cells manifest signs of quiescence and are arrested in the G0 phase under normal physical conditions; however, severe and/or repeated injuries could induce cell-cycle dysregulation, including cell-cycle arrest and overproliferation [7].

Cell-cycle dysregulation can be induced via injuries and other stimuli activated in diabetic milieus, such as hemodynamic changes, high glucose in blood circulation, merging of inflammatory cells, and various cytokines. These changes influence almost all kinds of kidney cell that activate a wide range of transduction pathways. Even though mild injuries activate proliferation of renal cells to compensate for cell loss and reinstate renal function, severe and/or repeated injuries could cause cell-cycle dysregulation, promoting glomerular and interstitial sclerosis and eventually renal fibrosis [7]. In glomeruli, podocytes with dysregulated cell cycles are activated and enter the cell cycle, yet are unable to complete the cycle and arrested in its M phase, finally leading to mitotic catastrophe and renal fibrosis. MCs with dysregulated cell cycles manifest as activation and overproliferation, resulting in excessive and persistent ECM depositing, which leads to glomerular sclerosis. GECs with dysregulated cell cycles are prone to overproliferation in the early stages of DKD, resulting in angiogenesis under regulation of podocytes via paracrine; however, this might lay the ground for glomerular sclerosis in later DKD with a decrease in VEGF-A. In renal tubules, TECs and fibroblasts can be converted to myofibroblasts via EMT and activation, respectively, which leads to excessive ECM acceleration, and TECs can also be arrested in the G2/M phase and undergo senescence, promoting renal fibrosis through secretion of profibrotic cytokines. Pericytes are another source of myofibroblasts and undergo differentiation with stimulation of the diabetic milieu. Inflammatory cells and factors are also crucial in the abovementioned processes, mediating cell overproliferation and renal fibrosis.

Evidence-based DKD management begins with a focus on hyperglycemia, blood pressure (BP), and albuminuria. Achieving and maintaining euglycemia are crucial for both T1DM and T2DM patients. The Systolic Blood Pressure Intervention Trial, 2015, demonstrated that among patients with high cardiovascular risk, lower target blood pressure (systolic BP < 120 vs. <140) was related to lower cardiovascular and all-cause mortality. Agents for blood-pressure controls, such as sodium–glucose cotransporter 2 inhibitor (SGLT2I) and glucagon-like peptide-1 receptor analog (GLP-1 RA), have been recently shown to control progression of DKD (Table 1) [72]. The role that cell-cycle dysregulation plays in progression of DKD leaves us with several treatment targets, and therapeutic strategies that are independent of their glycemia-control or BP-lowering effects. Metformin is reported to induce autophagy of MCs via the sirtuin 1- forkhead box protein O1 (SIRT1–FOXO1) autophagy axis and suppress high-glucose-induced MC proliferation, inflammation, and ECM expression [14]. It also mitigates podocyte loss, MC apoptosis, and tubular cell senescence via AMPK/mTOR regulation [73]. RAAS blockers, such as angiotensin-converting enzyme inhibitors (ACEIs), angiotensin-II receptor blockers (ARBs), mineralocorticoid antagonists, and direct renin inhibitors (DRIs), can reduce albuminuria and alleviate the ensuing TEC proliferation and interstitial inflammation and scarring. The cell-cycle arrest in DKD, however, has not yet been treated effectively.

In conclusion, cell-cycle dysregulation plays a crucial role in DKD and finally leads to renal fibrosis and end-stage renal disease (ESRD). Even though the specific mechanisms thereof remain to be fully understood and there are many other molecules and pathways left to be clarified, these findings have opened a window for new therapeutic strategies that could help mitigate development of DKD and prevent the outcome of ESRD.

## Figures and Tables

**Figure 1 ijms-24-02133-f001:**
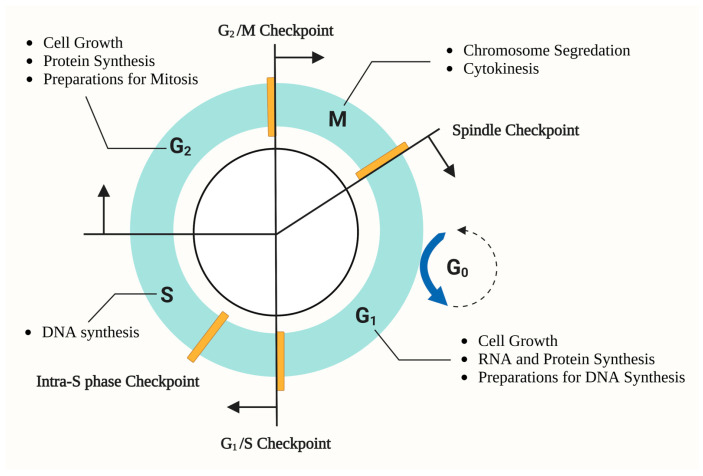
Features of the mammalian cell cycle.

**Figure 2 ijms-24-02133-f002:**
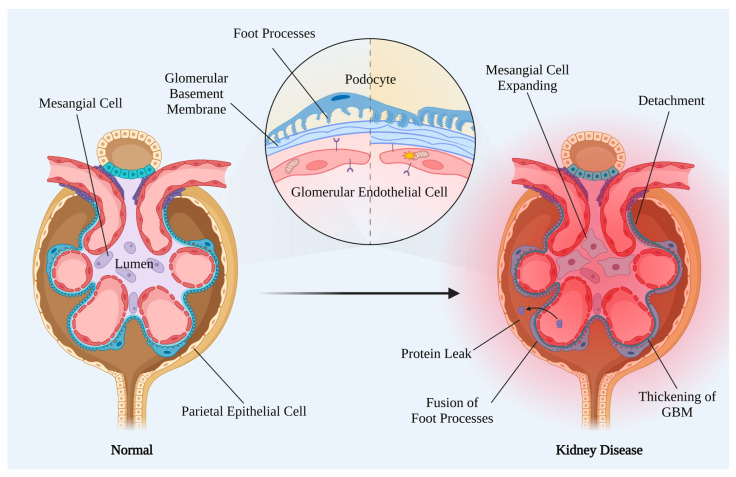
Features of normal and DKD glomerular filtration barriers. The FPs and fenestrated GECs are attached to the GBM, comprising a glomerular filtration barrier. In DKD, the glomerulus undergoes profound morphological changes, including thickening of the GBM and mesangial matrix expansion, occluding its capillaries with the extracellular matrix. Moreover, hemodynamic and metabolic milieu changes lead to FP fusion, podocyte detachment, GEC dysfunction, and the ensuing protein leakage thereof. FP: foot processes; GBM: glomerular basement membrane; GEC: glomerular endothelial cell; MC: mesangial cell; PEC: parietal epithelial cell.

**Figure 3 ijms-24-02133-f003:**
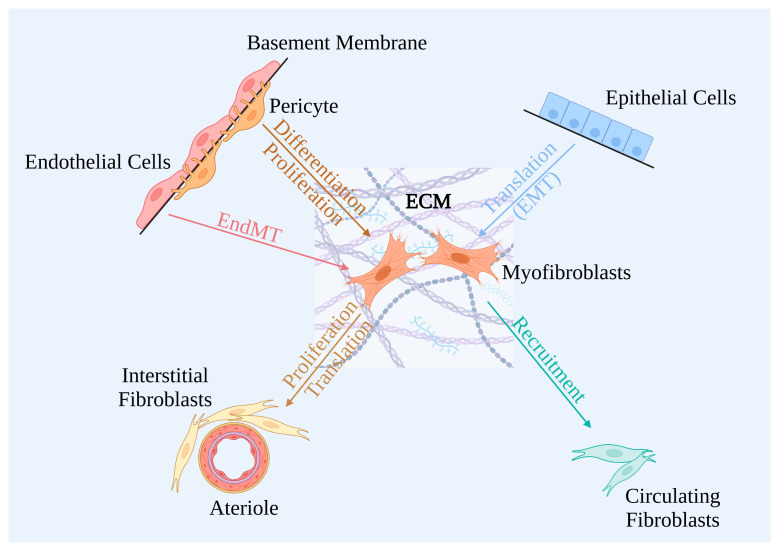
Features of myofibroblast origins.

**Table 1 ijms-24-02133-t001:** The evidence-based management of DKD.

Agents	Medicine	Mechanisms/Pathways	References
SGLT2Is	Empagliflozin	Reduction in high-glucose-induced oxidative stress and miR-21-dependent TRAF3IP2 induction and RECK suppression	doi:10.1016/j.cellsig.2019.109506 [46].
	Canagliflozin	Inhibition of SGLT-2	doi:10.1056/NEJMoa1611925 [74].
	Dapagliflozin		doi:10.1056/NEJMoa1812389 [75].
GLP-1 Ras	Liraglutide	Reduction in both oxidative stress and inflammation	doi:10.1038/ki.2013.427 [76].
	Semaglutide		
Metformin	/	Induction of MC autophagy via the SIRT1-FOXO1-autophagy axis	doi:10.1111/1440-1681.13120 [14].
RAAS blockers	ACEI	Reduction in albuminuria and alleviation of renal fibrosis	doi:10.1016/j.mce.2009.09.009 [48].
	ARB		
	mineralocorticoid antagonists		
	DRI		

## Data Availability

Not applicable.

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
