# Peer review of "Cell-Cycle Dysregulation in the Pathogenesis of Diabetic Kidney Disease: An Update"

_ijms, 2023, doi:10.3390/ijms24032133_

Round 1

Reviewer 1 Report

In this review article, Chinese authors describe the role of cell cycle dysfunction in diabetic kidney disease. The article is well presented, organised, and divided into many sections in which the authors analyse different parts of the nephron - podocytes, mesangial cells, glomerular endothelial cells, tubular epithelial cells, renal interstitium, and pericytes.

I have the following questions, recommendations and suggestions:

1.     I miss a more detailed description and discussion focusing on cell cycle dysregulation, which is the main goal of this article. 

2.     The term "diabetic nephrology" on page 5, row 177, 191 etc is a very unusual phrase that is not commonly used among nephrologists. Please correct. 

3.     On the page 5, row 180, for diabetic kidney disease the abbreviation DKD should be use.

Reviewer 2 Report

This is a well written, interesting manuscript on cell cycle dysregulation in the pathogenesis of diabetic kidney disease. Well sustained on updated bibliography it offers a nice and readable review of a complex set of mechanisms associated with diabetes kidney disease evolution and prognosis. However, some aspect can contribute with a better understanding on these complexities for the benefit of the readers:

11)      Authors mention: “The earliest observable glomerular change by light microscopy is diffuse mesangial expansion, which occurs as early as the fifth year after diabetes onset”. Even when the timing (fifth year after diabetes onset) is well accepted for patients with Type 1 diabetes, it is less clear in Type 2 diabetes patients. Some patients with type 2 diabetes display microscopic alterations at diagnosis and diabetes onset is difficult to establish in these patients. Even when most of the histological alterations are similar for both types of diabetes, some patterns may differ as well as general epidemiology and clinical evolution.

22)      The role of hyperglycemia on the development of cell cycle dysregulation is not as well describe as for other factors such as RAAS alterations. Is hyperglycemia a causally related factor for cell cycle dysregulation? Is this association clear enough? If yes, what would be the mechanisms? In general, the association between hyperglycemia and cell cycle alterations merits for a deeper discussion taking into consideration its clinical implications.

33)      Is the described cell cycle dysregulation associated with other cellular processes already described in diabetes kidney disease, such as autophagy, mitophagy, etc? A short section describing these associations may contribute with a better understanding of the critical integrated role of cell cycle alterations in the pathophysiology of diabetes kidney disease.

44)      Are there any pharmacological perspectives to discuss? The effects of some pharmacological agents are briefly described at some sections (e.g. SGLT2i). Are there newer targeted agents with a potential for disease modification being investigation at this moment? A table and a couple of paragraphs may be of help to understand the clinical implications of cell cycle modifications on DKD. 

3  

Round 2

Reviewer 2 Report

Adjustments to manuscript were satisfactorily done. All the recommended enhancements / modifications were satisfactorily included.